# Indirect Assessment of Skeletal Muscle Glycogen Content in Professional Soccer Players before and after a Match through a Non-Invasive Ultrasound Technology

**DOI:** 10.3390/nu12040971

**Published:** 2020-04-01

**Authors:** Iñigo San-Millán, John C. Hill, Julio Calleja-González

**Affiliations:** 1Department of Medicine, Division of Endocrinology, Metabolism and Diabetes, University of Colorado School of Medicine, Aurora, CO 80045, USA; 2Department of Human Physiology and Nutrition, University of Colorado, Colorado Springs, CO 80918, USA; 3Department of Family Medicine, University of Colorado School of Medicine, Aurora, CO 80045, USA; johnhill@ucdenver.edu; 4Laboratory of Analysis of Human Performance, Department of Physical Education and Sport, Faculty of Education, Sports Section, University of the Basque Country, 01007 Vitoria-Gazteiz, Spain

**Keywords:** MuscleSound, muscle glycogen, soccer, match

## Abstract

Skeletal muscle glycogen (SMG) stores in highly glycolytic activities regulate muscle contraction by controlling calcium release and uptake from sarcoplasmic reticulum, which could affect muscle contraction. Historically, the assessment of SMG was performed through invasive and non-practical muscle biopsies. In this study we have utilized a novel methodology to assess SMG through a non-invasive high-frequency ultrasound. Nine MLS professional soccer players (180.4 ± 5.9 cm; 72.4 ± 9.3 kg; 10.4% ± 0.7% body fat) participated. All followed the nutritional protocol 24 h before the official match as well as performing the same practice program the entire week leading to the match. The SMG decreased from 80 ± 8.6 to 63.9 ± 10.2; *p* = 0.005 on MuscleSound^®^ score (0–100) representing a 20% ± 10.4% decrease in muscle glycogen after match. Inter-individual differences in both starting glycogen content (65–90) and in percentage decrease in glycogen after the match (between 6.2% and 44.5%). Some players may not start the match with adequate SMG while others’ SMG decreased significantly throughout the game. Adequate pre-match SMG should be achieved during half-time and game-play in order to mitigate the decrease in glycogen. Further and more ample studies are needed before the application of this technology.

## 1. Introduction

Proper glycogen content plays a major role in athletic performance as has been described in the scientific literature for decades [1]. As shown through physiological measurements conducted during a soccer match, the bioenergetics of this particular sport are quite comprehensive and comprised of a high oxidative capacity, high glycolytic capacity as well as high reliance on the ATP/phosphocreatine breakdown/resynthesize pathway [2]. Due to the high glycolytic component, muscle glycogen content is relevant in soccer. In particular, the highest muscle glycogen synthesis rates have been reported when large amounts of carbohydrate (1.0–1.85 g/kg·h) are consumed immediately post-game and at 15–60-min intervals thereafter, for up to 5 h post-exercise [3].

A novel study related to muscle glycogen, led by Ørtenblad and Nielsen’s group, has opened new doors in understanding the potential key role of glycogen in skeletal muscle contraction through its regulation of calcium (Ca^2+^) release and uptake from the sarcoplasmic reticulum [4]. In this study, the authors observed that a decrease of ~25% of glycogen from the vastus lateralis muscle corresponded to ~10% decrease of Ca^2+^ release and uptake from the sarcoplasmic reticulum. Furthermore, the investigators described that the decrease in cytoplasmic free Ca^2+^ during repeated tetanic contractions is associated with the subcellular glycogen concentration [5]. Since intracellular Ca^2+^ is involved in muscle contractile force and fatigue [6], a decrease in muscle glycogen content could have important consequences for athletic performance.

Furthermore, Bromstrand and Saltin in 1999 concluded that a decrease in muscle glycogen storage lowered glycolytic capacity [7]. Specifically, in soccer, Krustrup et al. (2006) observed through muscle biopsies that the muscle glycogen concentration at the end of the match was reduced from 350 to 150 mmol/1 kg of dry body mass. Although the underlying mechanism behind a reduced exercise performance at the end of soccer games remains unclear [2], one potential explanation could be the decrease in glycogen stores below the required value of 200 mmol/1 kg of dry weight to maintain glycolytic rate. From the bioenergetics perspective, since the glycolytic contribution to soccer is crucial, a decrease in muscle glycogen stores could decrease glycolytic capacity and thus performance.

Due to the importance of skeletal muscle glycogen in sports performance, there is a need for monitoring skeletal muscle glycogen in a non-invasive manner [8], as well as for an analysis of the main components involved in glycogen depletion during a match, including frequency, duration, intensity and nutrition, which could contribute to the creation of individualized protocols for carbohydrate administration to soccer players. However, the measurement of skeletal muscle glycogen through a quick and non-invasive tool has remained elusive for many years, being only possible through invasive and non-practical muscle biopsies or expensive and impractical magnetic resonance techniques. Recently, our research group has developed and validated a novel methodology to assess muscle glycogen through high-frequency skeletal muscle ultrasound in a fast, portable and non-invasive manner [9]. This new device and technique present new possibilities to manage and individualize nutrition and recovery in athletes. To our knowledge, no previous studies have described the glycogen status in soccer players pre- and post-game using this novel technology. Therefore, the main aim of this pilot study was to describe the changes in skeletal muscle glycogen content in professional soccer players before and after an official soccer match through high frequency skeletal muscle ultrasound.

## 2. Methodology

### 2.1. Participants

Nine professional soccer players from Major League Soccer (MLS) in the US (Height: 180.4 ± 5.9 cm; Body mass: 72.4 ± 9.3 kg; Body fat: 10.4 ± 0.7%) from different positions (one goal keeper, three defenders, three midfielders, two attackers) participated in this study. All participants followed the nutritional protocol provided by the team for 24 h before the official match during the MLS as well as performing the same practice program during the entire week leading up to the competitive match supervised by the team. Right after the warm-up phase, all players ingested a drink containing 40 g of carbohydrates (CHO). Furthermore, at half-time, all players ingested another drink containing 40 g of CHO and another 25 g of CHO in a gel form. The total amount of CHO ingested after the warm-up (when first ultrasound scan was performed) plus the CHO ingested at half-time totaled 105 g (40 g after the warm-up (30 min before match) and 65 g of CHO ingested at half-time). A typical training day consisted of one 1.5 h session involving tactical and technical drills (50% of the practice), power and speed drills (25% of the practice), and interval sessions (25% of the practice).

This training program was repeated daily except on match days, when the match was performed, and on the day following the match which was a light technical training day. The experimental procedures, and associated risks and benefits, were explained to each player, and each player signed a written consent form before commencing participation. None of the players had any preexisting injuries prior to testing. This study was designed according to the Declaration of Helsinki (Fortaleza, 2013) and approved by the Internal Review Board of the University of Colorado (COMIRB Protocol number: 14-0711) (Table 1).

### 2.2. Procedures

The last meal they had was 2.5 h prior to the match. Players were required to arrive 1 h and 30 min before the start of the warm-up prior to the match. All scans were performed in the team’s locker room near to the soccer field. The warm-up included light jogging, sport-specific movements (changes of direction, jumps and shuffles), accelerations and active stretching. The first ultrasound scan was performed immediately before the warm-up and the second ultrasound scan was performed immediately after the match on all players who completed the full match (90 min). Only the players who played the entire game (*n* = 9) were scanned.

### 2.3. Muscle Glycogen Assessment

#### Pixel by Pixel Calculation to Establish the Glycogen Score Based on Ultrasound Echogenicity

MuscleSound^®^ technology (Denver, CO, USA) is based on skeletal muscle echogenicity from intracellular glycogen and water dynamics, as 1 g of water retains 3 of glycogen [9]. Briefly, the software can quickly process high-resolution DICOM images of specific muscles obtained from the high-frequency ultrasound to create a quantifiable score of muscle glycogen content based on the echogenicity of the high-frequency ultrasound. The software can identify the adipose tissue, connective tissue and muscle fascia in order to specifically isolate and crop the muscle assessed (Figure 1). The image is then changed to a binary (black and white) image [9]. The darker the image is, the more hypoechoic the ultrasound is and therefore the more glycogen-bound water is in the skeletal muscles. The whiter the image, the more hyperechoic the ultrasound is and the less glycogen-bound water is in the skeletal muscle fibers [9]. Adipose and connective tissues have the highest echogenicity. For those tissues, the pixel intensity of the image is 255. The remaining cropped and isolated muscle tissue has a pixel intensity between zero and 254. Once the adipose and connective tissues were isolated (Figure 1), the echogenicity of the image was translated to pixel intensity and the mean pixel intensity of the muscle was calculated, creating the glycogen score on a scale from zero to 254 of pixel intensity. This initial score from pixel intensity was then standardized to a scale from zero to 100 which was the scale used in the validation with muscle biopsy [9,10]. Furthermore, as Figure 1 shows, for aesthetics and user visuals, the white adipose and connective tissues were automatically stained a green color by the software detecting those tissues. Likewise, the pixel intensity of skeletal muscle echogenicity corresponding to the different intensity of white colors was stained a violet color.

It is worth noting that it is well known that muscle glycogen is stored in different pools within a same muscle and in different muscles according to different muscle fibers [5]. Therefore, a muscle biopsy which just collects ~1 to 2 cm^2^ cannot represent the glycogen store of an entire muscle but just the content of that specific biopsy tissue analyzed histologically. To clarify, in our previous validation study [9] we collected ~50 mg of muscle tissue from the rectus femoris muscle. We then correlated the muscle tissue glycogen content pre- and post-exercise through traditional histological quantification with the echogenicity of the image taken specifically from the exact same site as that from which the muscle biopsy using MuscleSound^®^ software was taken. To achieve this, we performed an ultrasound-guided muscle biopsy in order to identify and obtain the image of the specific site from where the biopsy was taken. This same methodology was also deployed by a different validation study by Nieman’s group [10]. Through this methodology both studies obtained strong correlations between pre- and post- exercise (r = 0.90, *p* < 0.001). Furthermore, a recent study has shown poor correlation between MuscleSound^®^ software and glycogen content inferred from muscle biopsy [11]. However, the methodology utilized in this study of the correlation was not adequate as it correlates the glycogen content from the muscle biopsy (1 to 2 cm^2^) with the software score of the entire muscle. To clarify again, since glycogen is stored in different pools within a same muscle and therefore not uniformly stored, technically it is not possible to correlate the glycogen content from a very small portion of a muscle (1 to 2 cm^2^) with the glycogen content of an entire muscle. Furthermore, this aforementioned study was performed through traditional “blind” biopsy and therefore it was not possible to collect an image of the specific muscle site from where the biopsy was taken.

Moreover, a recent study by Shiose et al. demonstrated that water bound to glycogen is intracellular [12], and muscle glycogen depletion did not alter segmental extracellular and intracellular water distribution as measured through bioimpedance spectroscopy [13]. Therefore, the use of high-frequency ultrasound to study the echogenicity of muscle glycogen depletion should not be altered. Further, in our validation study [9], as well as Nieman’s study [10], participants were forced to drink water ad libitum during the exercise protocol, resulting in a decrease in body weight of <2%, which is not considered clinical dehydration.

In this study, we indirectly assessed muscle glycogen in the muscle before and after a game through high frequency ultrasound scans using the described technology. Muscle ultrasound scans were obtained from the rectus femoris (RF) muscle pre- and post-exercise. Muscle scans were performed with a 12 MHz linear transducer at a gain of 50 for all subjects, and a standard diagnostic high-resolution ultrasound machine (GE LOGIQe (GE Healthcare, Milwaukee, WI, USA)). Two ultrasound scans were performed on each player each time, both before and after the game. The software calculated the mean score of the two scans (score 0–100). The first measurement was performed 10 min before the warm-up and the second measurement was performed 5 min after the end of the game when players returned to the locker room. All subjects were in a supine position on a massage table to perform the ultrasound scan. Due to the rapidness of the ultrasound the procedure, the scans for each player took about 30 s to 1 min, so the entire team was scanned within 10 min.

### 2.4. Statistical Analyses

Data are presented as means and standard errors or standard deviations. We applied the Shapiro–Wilk Test which requires the sample size to be between three and 50 [14]. Differences from pre to post were assessed using a paired Student’s *t* test to decide parametric analysis. Group differences at pre-match and the percentage change of the outcome variables from post-match were calculated as _ (%): ((Pre – Post)/post) × 100. The effect sizes between participants were calculated using partial square eta (η2p). Since this measure is likely to overestimate the effect sizes, the values were interpreted according to [15] which was indicated as having no effect if 0 ≤ η2p < 0.05; a minimum effect if 0.05 ≤ η2p < 0.26; a moderate effect if 0.26 ≤ η2p < 0.64; and a strong effect if η2p ≥ 0.64. Analyses were performed using IBM SPSS Statistics for Windows, version 21.0 (IBM Corp., Armonk, NY, USA). Statistical significance was set at *p* < 0.005.

## 3. Results

After the 90 min of the match, the average rectus femoris glycogen content decreased from 80 ± 8.6 points before the game (MuscleSound^®^ score 100 points) to 63.9 ± 10.2 points after the game (*p* = 0.005; η2p = 1). Each player’s glycogen content decreased at various rates, −6.2 to −44.5%; team average = 19.98% (Table 2). The highest decreases in glycogen were shown in forwards and midfielders (27.6% ± 13.8%) relative to defense players (11.8% ± 6.9%). As expected, the lowest decrease in glycogen was observed in the goal keeper with a modest 6.2% decrease in glycogen (Table 2 and Figure 2).

## 4. Discussion

This pilot study was designed to evaluate the changes in skeletal muscle glycogen content in professional soccer players from the MLS after an official soccer game, using a new non-invasive device which could offer novel possibilities to individualize nutritional protocols for soccer players. Muscle biopsies to assess glycogen content have been performed for several decades. However, due to the invasive nature of muscle biopsies, the assessment of muscle glycogen through non-invasive methodologies has remained elusive. Further, the use of muscle biopsies is almost always limited to a very small biopsy of the rectus femoris muscle which, as mentioned in the Methods section, does not necessarily represent the glycogen content of the entire muscle and doesn’t differentiate glycogen content among muscles. Our novel methodology allows for the indirect assessment of the muscle glycogen content of the entire muscle and offers the possibility to assess multiple muscle groups. The main findings in this study were that the average rectus femoris glycogen content decreased from 80 ± 8.6 points (MuscleSound^®^ score points) before the game to 63.9 ± 10.2 points after the game (*p* = 0.005), with a large effect size, representing a ~20% decrease in muscle glycogen content from the rectus femoris muscle. Furthermore, all players decreased glycogen content at different rates (−6.2% to −44.5%), which showed important differences in glycogen depletion rates among players. According to previous studies muscle glycogen decreased from 449 ± 23 to 255 ± 22 mmol/1 kg of dry weight during the game (*p* < 0.05), with 47% ± 7% of the muscle fibers being completely or almost empty of glycogen after the game [2]. However, there was no CHO supplementation reported in this study at half-time, while it is worth noting that the players in this study ingested 105g of CHO between the end of the warm-up (first scan) and the end of the game (second scan). Even with this nutritional protocol we observed significant decreases in glycogen in a few players. Therefore, we believe that proper and individualized nutritional protocols during half-time as well as throughout the game should be stressed and further studied in order to mitigate such significant variabilities in glycogen depletion among players. Although in this pilot study we did not measure running distance or intensities covered throughout the game through GPS-based technology, further studies measuring these parameters should shed more light into individual and team glycogen depletion patterns during a soccer game. Further studies are encouraged to include metabolic data from each player in order to understand and correlate the data from the indirect assessment of skeletal muscle glycogen with the metabolic parameters of each player. Certain players are more glycolytic and others also have different metabolic flexibility, being more efficient metabolically speaking at utilizing fat and CHO, mainly due to an increased mitochondrial function and glycolytic capacity as we have recently shown in populations of different fitness status [16].

Comprehension of the underlying mechanisms behind the onset of fatigue towards the end of the game in soccer matches is arguably more critical, given that this type of fatigue will potentially affect the majority of soccer players [17]. Elite soccer players tend to decrease performance during the second part of the game [18]. Accumulating evidence attributes fatigue in the latter stages of the game to a glycogen depleted [19]. Our study showed that decreases in glycogen content during the game were up to 44% in some players, which could represent an important and determinant decrease in performance. Moreover, glycogen content regulates Ca^++^ release and uptake from sarcoplasmic reticulum, and therefore decreased muscle glycogen stores could result in a significant decrease in muscle contraction capacity, which could have important negative effects on performance.

Furthermore, it was observed that muscle glycogen content prior to the game in two of the nine players studied was 65 MuscleSound^®^ points which was ~20% lower than the average of the team (80 points). This shows that some players may not have followed the nutritional recommendations given by the team before a game, which exposes the need for proper and individualized CHO intake prior to a soccer match, given that the activity profile and physiological demands on a player are determined by his/her positional role on the team [17], with the most physically-demanding playing positions having been shown to suffer a reduction in match performance in the second half in the Premier League [20].

In light of our findings, the assessment of skeletal muscle glycogen in soccer players could be beneficial in individualizing nutrition during the game, which could elicit important consequences for performance. Finally, future studies in combination with GPS-based tracking devices could allow a more precise identification of both individual and team fatigue characteristics, which lead to decreases in performance throughout the match, and such studies could thus enhance the analytic tools of any football team and lead to improved performance.

## 5. Conclusions

The novel methodology of using MuscleSound^®^ to indirectly assess skeletal muscle content showed in this pilot study that during a soccer game, skeletal muscle glycogen decreases at various rates among players, which could be attributable to different factors such as metabolic differences in substrate utilization, game demand and/or differences in CHO replacement during the game. Additionally, important differences were identified in pre-match glycogen content between two players who probably did not have an adequate CHO intake pre-match, which shows the necessity of proper and individualized CHO intake leading to a soccer match.

## 6. Limitations

The nature of this pilot study was to explore the possibilities that a new methodology could offer to indirectly assess muscle glycogen content in a non-invasive and rapid manner in order to individualize nutrition in soccer players. Further studies are needed to validate the possibilities of this methodology.

Probably one of the most important limitations of this pilot study was its small sample size. However, the majority of the traditional and seminal muscle biopsy studies were also comprised of a small *n* (some smaller than our study). Furthermore, professional soccer is only performed with 11 players per team and the chances are that, due to substitutions during the game, less than 11 players will play the entire game (nine players in our case).

On the other hand, in this pilot study there was a CHO ingestion by the players and (due to the nature of the professional competitive game) no controlled conditions vs. no CHO ingestion.

Furthermore, for this pilot study, we did not deploy GPS-based tracking methods to assess multiple parameters like speed, distance covered, sprints or accelerations per player. Due to the large inter-game variability regarding external load in soccer, further studies need to be performed encompassing GPS-based tracking methods as well as individual metabolic capabilities to further determine the specific influence of the external workload for each player on their glycogen content during a match.

## Figures and Tables

**Figure 1 nutrients-12-00971-f001:**
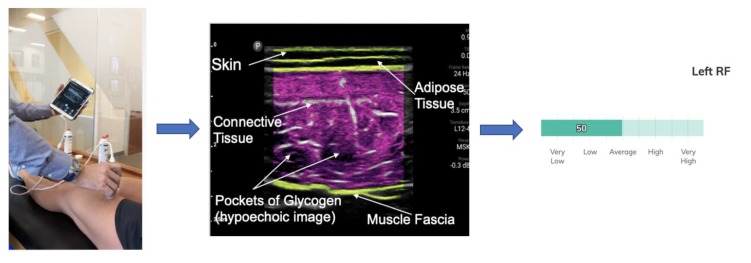
Muscle scan of rectus femoris (RF) being captured, followed by MuscleSound^®^ cropping and staining based on echogenicity of the muscle, resulting in a score measured by MuscleSound^®^ software, according to Hill and San-Millan [9].

**Figure 2 nutrients-12-00971-f002:**
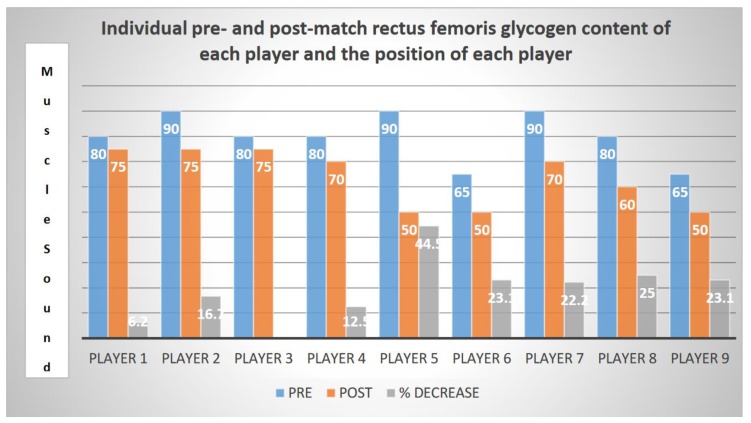
Individual pre- and post-match rectus femoris glycogen content of each player and the position of each player.

**Table 1 nutrients-12-00971-t001:** Nutritional and carbohydrate protocol followed before the game and at half-time.

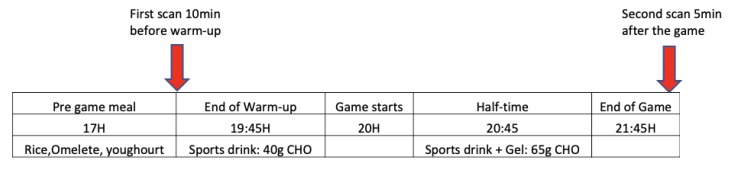

No carbohydrate (CHO) consumption was allowed after the game until the ultrasound scan was completed.

**Table 2 nutrients-12-00971-t002:** Individual pre- and post-match rectus femoris glycogen content of each player and the position of each player.

Player	MuscleSound Score Pre	MuscleSound Score Post	% Decrease	Effect Size	Position
				η2p	
Player 1	80	75	6.2		Goal Keeper
Player 2	90	75	16.7		Defense
Player 3	80	75	6.2		Defense Center
Player 9	80	70	12.5		Defense
Player 5	90	50	44.5		Midfielder
Player 6	65	50	23.1		Midfielder
Player 7	90	70	22.2		Midfielder
Player 8	80	60	25		Forward
Player 9	65	50	23.1		Forward
Team Average	80 ± 8.6	63.9 ± 10.19	20 ± 10.4	1	*p* = 0.005

Two players showed up to the game with a score of 60 which was 25% lower than the average of the team (80). Furthermore, these two players also showed the lowest post-match glycogen levels (Table 2). It is also worth noting that all players received at half-time 65 g of CHO (40 g within a drink and 25 g in a sports gel form).

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
