# Peer review of "Indirect Assessment of Skeletal Muscle Glycogen Content in Professional Soccer Players before and after a Match through a Non-Invasive Ultrasound Technology"

_nutrients, 2020, doi:10.3390/nu12040971_

Round 1
Reviewer 1 Report
Summary:
In the present study, San-Millan, Hill, and Calleja-Gonzalez evaluated glycogen depletion in professional soccer players using the novel MuscleSound technique. This technique is a non-invasive methodology using ultrasound to determine glycogen content in muscle. The authors measured glycogen content in the rectus femoris of 9 soccer players pre and post competition. As expected, glycogen content significantly decreased with a greater increase in offensive players compared to defensive players. The authors conclude that nutritional strategies that prevent glycogen depletion should be addressed in athletes.
Comments, Concerns, and Suggestions:
- The technology used for data collection is quite interesting and potentially useful. However, the data is limited to one table of information, which highlights glycogen change in one muscle, after one competition.
- Did the authors measure additional muscles? Is there uniform glycogen depletion in multiple lower extremity muscles?
- The findings would be strengthened by comparing the glycogen content with some variables of performance.
- The significance, importance, and the potential impact of the current work is not clear. Essentially, the major finding states that glycogen depletes in an intensity-dependent manner. This is already known. The reviewer assumes the major point of the paper is to use the MuscleSound technology as a novel data acquisition tool. The previous publication appears to have already done this in a more rigorous manner.
- The authors make the statement that some players did not have adequate glycogen stores. How was the adequacy of glycogen stores determined? Is there a genetic component to glycogen stores? If performance or other variables were not assessed, how can the starting and ending glycogen stores be defined as inadequate?
Author Response
Thank you very much for your comments.
REVIEWER 1: The technology used for data collection is quite interesting and potentially useful. However, the data is limited to one table of information, which highlights glycogen change in one muscle, after one competition.
AUTHORS: Thanks so much for your consideration. In fact, this is the main purpose of our study and the strength of the new non-invasive device. This a preliminary and pilot study which shows the possible applications of this methodology. However, as the Reviewer points out, the data is limited and further data with more muscle groups and soccer teams is necessary.
It is also worth noting that all the historic data that we have from the scientific literature from muscle biopsies is also from one same muscle and the “n” has always been very small.
REVIEWER 1: Did the authors measure additional muscles? Is there uniform glycogen depletion in multiple lower extremity muscles?
AUTHORS: We thank the reviewer for this important comment. We have started to also measure hamstrings muscles in addition to rectus femoris and we are seeing already significant differences. We still don’t know why but it is worth exploring this further. As the Reviewer knows and we point out in the manuscript, glycogen is not stored uniformly across different muscles and even within a same muscle. Therefore, for future studies, we believe that the inclusion of more muscles is necessary. The aim of our pilot study was to show the “proof of concept” which could open the door to further studies and application if this technology to improve monitoring as well as individualization of nutritional protocols in soccer players.
REVIEWER 1: The findings would be strengthened by comparing the glycogen content with some variables of performance.
AUTHORS: The Reviewer is absolutely right. As we point out in the limitations section of the manuscript we agree that comparing glycogen pattern variations with performance should be necessary in further studies. Currently we are starting to design a study to incorporate GPS-based tracking technology in order to establish some performance correlations.
REVIEWER 1: The significance, importance, and the potential impact of the current work is not clear. Essentially, the major finding states that glycogen depletes in an intensity-dependent manner. This is already known. The reviewer assumes the major point of the paper is to use the MuscleSound technology as a novel data acquisition tool. The previous publication appears to have already done this in a more rigorous manner.
AUTHORS: We thank the Reviewer for this comment. The main purpose we try to express through this manuscript was the utilization of this methodology in a field and real situation manner like a soccer game. Our previous validation (Hill and San Millan, 2014) was, as the reviewer points out, under rigorous laboratory conditions and with cyclists. This pilot study is pursuing the possible application in real game situations with soccer players.
REVIEWER 1: The authors make the statement that some players did not have adequate glycogen stores. How was the adequacy of glycogen stores determined? Is there a genetic component to glycogen stores? If performance or other variables were not assessed, how can the starting and ending glycogen stores be defined as inadequate?
AUTHORS: The Reviewer makes a very valid and good point and we appreciate the input. In fact, and due to the nature of this pilot study we don’t know yet which levels of glycogen are normal and which are not as more studies need to be performed. However, since the scale from the software is 0-100, we assume that those athletes with a score in the 80’s-90’s should have optimum levels whereas those players with scores in the 50’s may not have optimal levels. Again, this is arbitrary and further studies are needed. We have added this comment in the manuscript in the methods section explaining the score system. We hope the Reviewer is satisfied with that inclusion.
Reviewer 2 Report
Abstract:
The abstract has more than 200 words, which is not in accordance with the rules.
When referring: "representing 20 ± 10.4% decreased in muscle glycogen after match. Inter-individual differences in both starting glycogen content (65-90 in MuscleSound® score) as well as in percentage decrease in glycogen after the match, 6.2% to 44.5% were observed." Did they all start with the same glycogen level?
Key words:
Only soccer appears on the MeSH
Introduction:
The reference ".... validated by Nieman et al." does not correspond to bibliographic reference number 5.
Methods:
"... our research group has developed and validated a new methodology to quickly assess muscle glycogen using high-frequency skeletal muscle ultrasound ..." How many individuals has your group validated?
MuscleSound® score (0-100)- clarify which score refers.
In the analysis on echogenicity, the authors should mention if the percentage of gains was equal in the analysis of all.
A description of the examination protocol should be improved:, how many measurements did you take, whether the mean value or higher was used, the patient's position ....
Results:
Was the reported reduction / variation in skeletal muscle glycogen related or did it take into account the fat and body mass of each individual?
Conclusions:
Do the authors not consider it important to mention that they had different positions and, consequently, different levels of effort?
The authors refer only to individual metabolism in the last paragraph of the article. Shouldn't individual metabolism be taken into account?
Bibliography:
It must be revised and written according to the magazine's rules: The are references with letters in the wrong format. For example, the first reference is in italics.
There is some inconsistency in the way references are presented. For example, the 2nd and 3rd reference have different scores
Author Response
Thank for your comments.
REVIEWER 2: The abstract has more than 200 words, which is not in accordance with the rules.
AUTHORS: Thanks so much for this detail. We have reduced the number of words based on guidelines. (YELOW COLOUR IN THE TEXT)
Skeletal muscle glycogen (SMG) stores in highly glycolytic activities regulate muscle contraction by controlling calcium release and uptake from sarcoplasmic reticulum which could affect muscle contraction. Historically, the assessment of SMG was performed through invasive and non-practical muscle biopsies. In this study we have utilized a novel methodology to assess SMG through a non-invasive high-frequency ultrasound. Nine MSL professional soccer players (180.4 ± 5.9 cm; 72.4 ± 9.3 kg; 10.4 ± 0.7 % body fat) participated. All followed the nutritional protocol 24-h before the official match as well as performed the same practice program the entire week leading to the match. The SMG decreased from 80 ± 8.6 to 63.9 ± 10.2; p= 0.005 on MuscleSound® score (0-100) representing 20 ± 10.4% decreased in muscle glycogen after match. Inter-individual differences in both starting glycogen content (65-90) and in percentage decrease in glycogen after match (6.2% to 44.5%). Some players may not start the match with adequate SMG while others decreased significantly the amount of SMG throughout the game. Adequate pre-match SMG should be achieved during half-time and game in order to mitigate the decrease in glycogen. Further and more ample studies are needed towards the application of this technology.
REVIEWER 2: When referring: "representing 20 ± 10.4% decreased in muscle glycogen after match. Inter-individual differences in both starting glycogen content (65-90 in MuscleSound® score) as well as in percentage decrease in glycogen after the match, 6.2% to 44.5% were observed." Did they all start with the same glycogen level?
AUTHORS: Thank you for the question. As we show in the data, the starting glycogen was between 65 and 90 in the 0-100 scale. Players six and nine started with a glycogen score of 65 while players two, five and seven started with a glycogen concentration of 90.
REVIEWER 2: Key words: Only soccer appears on the MeSH
AUTHORS: Thanks, you, we have included the “soccer Word” in the Key Words and in the title
Keywords: MuscleSound; muscle glycogen; soccer; match
Title: Indirect Assessment of Skeletal Muscle Glycogen Content in Professional Soccer Players before and after a Match through a non-invasive ultrasound technology.
REVIEWER 2: Introduction: The reference ".... validated by Nieman et al." does not correspond to bibliographic reference number 5.
AUTHORS: Thank you for the observation. We appreciate it. We have deleted this sentence in the introduction part (YELOW COLOR IN THE TEXT)
REVIEWER 2: Methods: "... our research group has developed and validated a new methodology to quickly assess muscle glycogen using high-frequency skeletal muscle ultrasound ..." How many individuals has your group validated?
AUTHORS: This article “Hill JC, Millán IS. Phys Sportsmed. 2014 Sep;42(3):45-52. doi: 10.3810/psm.2014.09.2075”, was published in 2014, and we used Twenty-two male competitive cyclists.
REVIEWER 2: MuscleSound® score (0-100)- clarify which score refers.
AUTHORS: MuscleSound software can quickly process high-resolution DICOM images of specific muscles obtained from the high-frequency ultrasound to create a quantifiable score of muscle glycogen content based on the echogenicity of the high-frequency ultrasound. The software can identify the adipose tissue, connective tissue and muscle fascia in order to specifically isolate and crop the muscle assessed (Fig-1). The image was then changed to a binary (black/white) image (Hill and San Millan, 2014). The darker the image is, the more hypoechoic and therefore the more glycogen-bound water in skeletal muscles. The whiter the image, the more hyperechoic the ultrasound is and the less glycogen-bound water in skeletal muscle fibers (Hill and San Millan, 2014). Adipose and connective tissues have the highest echogenic intensity. For those tissues, pixel intensity of the image is 255. The remaining, cropped and isolated muscle tissue has a pixel intensity between 0 and 254. Once the adipose and connective tissues were isolated (image 1) the echogenicity of the image was translated to pixel intensity and the mean pixel intensity of the muscle was calculated, creating the glycogen score on a scale from 0-254 of pixel intensity. This initial score from pixel intensity was then standardized to a scale from 0-100 which was the scale used in the validation with muscle biopsy (Hill and San Millan, 2014; Nieman et al, 2015). Furthermore, as figure 1 shows, for esthetics and user’s visual, the white adipose and connective tissues were automatically stained into green color by the software detecting those tissues. Likewise, the pixel intensity of skeletal muscle echogenicity corresponding to the different intensity of white colors was stained into violate color.
We have incorporated this paragraph into the manuscript for a better understanding. We thank the Reviewer for this appreciation.
REVIEWER 2: In the analysis on echogenicity, the authors should mention if the percentage of gains was equal in the analysis of all.
AUTHORS: We thank the Reviewer for this appreciation as well. The gain was never charged in any of the assessments as it must remain constant in order to provide the same echogenicity as the reviewer very well appreciates. We have also added this correction to the manuscript.
REVIEWER 2: A description of the examination protocol should be improved: how many measurements did you take, whether the mean value or higher was used, the patient's position ....
AUTHORS: We thank the reviewer again for this important appreciation. We performed two scans per player of the rectus femoris muscle. The software calculates the mean of the two scans. Furthermore, the software has a function where it can detect possible errors from the user like wrong placement of the probe which can cause a hypoechoic lateral artifact of the image or an artifact resulting in hyperechoic image from excessive pressure of the probe. If during the cropping process of muscle isolation there is any mistake, the software will warn the user who can visualize that image, delete it or re-do the scan.
All subjects were in a supine position on a massage table. We appreciate the reviewer noting and suggesting this point.
REVIEWER 2: Results: Was the reported reduction / variation in skeletal muscle glycogen related or did it take into account the fat and body mass of each individual?
AUTHORS: The software assesses the echogenicity of the entire muscle and returns a score which is not related to fat and body mass. We reported the average body fat of the players 10.4 ± 0.7 %, thus very little variation among subjects. However, the body mass’ variation is higher (72.4 ± 9.3 kg). The reviewer makes a very good point and strengthens the need to incorporate more parameters and evolve the software towards some form of quantification of a ratio between glycogen score/body mass for example as well as to perform further studies as we stress with this pilot study. Empirically, we have observed that elite athletes tend to have ~25% higher scores than recreational athletes probably because of increased lean muscle mass and also a higher glycogen capacity which has also been observed in the literature. With ICU patients we observed in a pilot study, scores between 0-15 (San Millan et al, 2015). These patients are characterized by increased glucose oxidation rates (2-3 times that of resting conditions in healthy individual’s) as well as muscle cachexia and proteolysis, possibly for gluconeogenic purposes due to decreased glycogen content to meet the necessary glycose demand from ICU patients for healing purposes. We are conducting a large clinical study at Duke University School of Medicine to establish different rations regarding glycogen and body mass in ICU patients. Furthermore, we have a manuscript under review with a study performed at MD Anderson Cancer Center where applying this methodology we also observe a decrease of about 50% in glycogen scores in cachectic cancer patients in comparison with non-cachectic cancer patients. We believe that this is due to the Warburg Effect characteristic of cancer, where both glucose utilization and cachexia are directly related to the severity and aggressiveness of solid tumors. For further reading about the Warburg Effect in cancer, if Reviewer is interested I would refer to our recent publication in Carcinogenesis (San-Millan and Brooks, 2017).
REVIEWER 2: Conclusions: Do the authors not consider it important to mention that they had different positions and, consequently, different levels of effort?
AUTHORS: Yes, we totally agree and as this is a pilot study, this is the main reason why, in the limitations paragraph, we have included the next sentence
“Due to the large inter-game variability regarding external load in soccer, further studies need to be performed including GPS-based tracking methods to further determine the specific influence of the external workload for each player on the glycogen content during a match”
REVIEWER 2: The authors refer only to individual metabolism in the last paragraph of the article. Shouldn't individual metabolism be taken into account?
AUTHORS: We thank the reviewer for this point. Certain players are more glycolytic and others also have different metabolic flexibility being more efficient metabolically speaking at utilizing fat and CHO, mainly due to an increased mitochondrial function and glycolytic capacity. Therefore, without a doubt, we believe that glycogen utilization during a game should be affected. However, metabolic measurements and specifically substrates utilizations must be done in a laboratory with metabolic carts and lactate testing. We have recently published an article describing an indirect methodology to assess metabolic flexibility and mitochondrial function in populations with different fitness levels (San-Millan and Brooks, 2018).
We thank the Reviewer for appreciating this great point about metabolism.
We have included in the manuscript a more detailed explanation about the individual metabolism among different individuals. We hope the Reviewer is pleased with it.
REVIEWER 2: Bibliography: It must be revised and written according to the magazine's rules: The are references with letters in the wrong format. For example, the first reference is in italics.
AUTHORS: Thanks so much. We have rewritten the references based on journal guidelines
REVIEWER 2: There is some inconsistency in the way references are presented. For example, the 2nd and 3rd reference have different scores
AUTHORS: Thanks so much. We have rewritten the references based on journal guidelines
Reviewer 3 Report
General comment
The authors performed a pilot study to investigate the changes in skeletal muscle glycogen content in professional soccer players before and after an official soccer match through high frequency skeletal muscle ultrasound. Results revealed that skeletal muscle glycogen levels as shown by MuscleSound® score decreased after soccer match while showing inter-individual differences probably due to differences of positions and nutritional conditions. The viewpoint of this study and the employment of professional soccer players and MuscleSound® are very interesting and valuable for high performance sports fields, but experimental design, sample size, expression of methods and results, and discussion/conclusion are seemed immature.
Minor comments
- Could the authors write the validity and accuracy of muscle glycogen detection by MuscleSound® much thoroughly?
- When did the authors measure muscle glycogen levels? Could the authors write more detailed information, for example how many minutes before and after match?
- Could the authors show a table for more details of nutritional protocols (meals and drinks) before and during soccer match not only CHO content but also all components?
- Did the authors measure the lactate, some glycolytic hormones, and some inflammation markers in blood or saliva? These factors can be supporting evidence of muscle glycogenolysis.
- Could the authors explain more detail of Figure 1 on the figure legends, for example which color means what?
- Authors mentioned “Data are presented as means and standard errors or standard deviations.” However, readers cannot understand which nuimbers standard errors or standard deviations are.
- Could the authors show some graph of results? Graphs, rather than a table, can be helpful for readers’ better understanding.
- Could the authors analyze the difference between positions by adding several participants to each position (Goal keepers, Defenders, Midfielders, Forwards)?
- Could the authors discuss the physical activity levels on each position during soccer match according to previous studies for example using GPS-tracking?
- Could the authors propose much concrete example of “proper CHO intake” based on the present study?
Author Response
Thank you very much for your comments.
REVIWERE 3: General comment: The authors performed a pilot study to investigate the changes in skeletal muscle glycogen content in professional soccer players before and after an official soccer match through high frequency skeletal muscle ultrasound. Results revealed that skeletal muscle glycogen levels as shown by MuscleSound® score decreased after soccer match while showing inter-individual differences probably due to differences of positions and nutritional conditions. The viewpoint of this study and the employment of professional soccer players and MuscleSound® are very interesting and valuable for high performance sports fields, but experimental design, sample size, expression of methods and results, and discussion/conclusion are seemed immature.
AUTHORS: We thank the Reviewer for his/her feedback. We agree that we the sample size is small and that also many more studies need to be done. That’s why we decided with a small pilot stud. At the end of the day, we have spent decades speaking, listening, reading and writing about the importance of glycogen with studies done with a very small sample size of five or ten subjects and not under real exercising conditions or competition of professional sports. I hope the reviewer understands the nature of our novel pilot study as we are the first ones admitting the limitations and encouraging others to continue on with further studies.
REVIEWER 3: Could the authors write the validity and accuracy of muscle glycogen detection by MuscleSound® much thoroughly?
AUTHORS: Throughout the manuscript we refer to the original validations (Hill and San Millan, 2014; Nieman et al, 2015).
We have added a paragraph to explain in more detail the technology “
Hill JC, Millán IS. Validation of musculoskeletal ultrasound to assess and quantify muscle glycogen content. A novel approach. Phys Sportsmed. 2014 Sep;42(3):45-52. doi: 10.3810/psm.2014.09.2075.
REVIEWER 3: When did the authors measure muscle glycogen levels? Could the authors write more detailed information, for example how many minutes before and after match?
AUTHORS: We thank the reviewer for this point. The first assessment was performed 10 minutes before the warm-up and the second 5 minutes after the game as soon as the players returned to the locker room. We have added this specification to the manuscript.
REVIEWER 3: Could the authors show a table for more details of nutritional protocols (meals and drinks) before and during soccer match not only CHO content but also all components?
AUTHORS: We thank the reviewer for this advice. We have added it as suggested.
REVIEWER 3: Did the authors measure the lactate, some glycolytic hormones, and some inflammation markers in blood or saliva? These factors can be supporting evidence of muscle glycogenolysis.
AUTHORS: We thank the important points that the Reviewer brings up. We have added a paragraph on the possible differences in metabolic responses among players based on metabolic flexibility and substrate utilization.
Certain players are more glycolytic and others also have different metabolic flexibility being more efficient metabolically speaking at utilizing fat and CHO, mainly due to an increased mitochondrial function and glycolytic capacity. Therefore, without a doubt, we believe that glycogen utilization during a game should be affected. However, metabolic measurements and specifically substrates utilizations must be done in a laboratory with metabolic carts and lactate testing. We have recently published an article describing an indirect methodology to assess metabolic flexibility and mitochondrial function in populations with different fitness levels (San-Millan and Brooks, 2018).
Furthermore, we have been working and developed modern techniques to assess the metabolic responses to exercise through metabolomics. Currently, we have a manuscript under review using metabolomics with Tour de France level cyclists. Through metabolomics, we have found important differences in multiple metabolic pathways involved in bioenergetics between cyclists at the highest level in the world. Through metabolomics we can measure about 2.000 parameters with a small blood sample. In our study under review we have measured glycolytic, oxidative, fatty acid, amino acids and pentose phosphate pathways. We can also measure multiple pathways involved in adrenergic activation or inflammation. Now that we have been able to develop and start validating these techniques, the addition to indirect glycogen assessment should shed new light and offer a deeper understanding on the metabolic responses to sports like soccer as well as the individual nutritional demands. This is the intention after this pilot study.
REVIEWER 3: Could the authors explain more detail of Figure 1 on the figure legends, for example which color means what?
AUTHORS: We thank the Reviewer for the feedback. In addition to a more elaborated paragraph describing our methodology as the reviewer suggested, we have added more detail to Figure 1. I hope the Reviewer is satisfied with these changes.
REVIEWER 3: Authors mentioned “Data are presented as means and standard errors or standard deviations.” However, readers cannot understand which numbers standard errors or standard deviations are.
AUTHORS: Data are presented as means and standard deviations.
REVIEWER 3: Could the authors show some graph of results? Graphs, rather than a table, can be helpful for readers’ better understanding.
AUTHORS: According your suggestions, we have included 1 table and 1 graph
REVIEWER 3: Could the authors analyze the difference between positions by adding several participants to each position (Goal keepers, Defenders, Midfielders, Forwards)?
AUTHORS: We thank the Reviewer for the comment. This is one of the limitations of our pilot study as we describe in the limitations section of the manuscript. Precisely, the nature and intention of this pilot study is to spark the interest in this methodology as a possible manner to provide feedback for coaches, scientists and athletes on individualized nutrition and monitoring of nutritional status. As addressed in the limitations section, it is necessary to perform more studies with a bigger number of athletes and teams. Certainly, the addition of a high number of players in the same position will provide us with more information on the behavior of glycogen dynamics during a soccer game. We believe that this would be necessary to do with many teams and players which should be a second step after this pilot study.
REVIEWER 3: Could the authors discuss the physical activity levels on each position during soccer match according to previous studies for example using GPS-tracking?
AUTHORS: This a preliminary data report, “Due to the large inter-game variability regarding external load in soccer, further studies need to be performed including GPS-based tracking methods to further determine the specific influence of the external workload for each player on the glycogen content during a match”
REVIEWER 3: Could the authors propose much concrete example of “proper CHO intake” based on the present study?
AUTHORS: We thank the Reviewer for the comment. We want to be cautious about providing “much concrete” nutritional conclusions on what the ideal nutrition should be before or during a soccer game. That is why this pilot study should lead to further and larger studies in order to understand in more detail the exact nutritional demands of soccer players in order to be able tom provide more concrete recommendations. We have rephrased our statements regarding “proper CHO intake” in case it can seem confusing. We hope the Reviewer understands our point.
Round 2
Reviewer 1 Report
The authors have responded adequately to the comments given the limitations of the study.
Reviewer 3 Report
The authors responded well to reviewer's comments.